# ICP-MS Multi-Elemental Analysis of the Human Meninges Collected from Sudden Death Victims in South-Eastern Poland

**DOI:** 10.3390/molecules27061911

**Published:** 2022-03-15

**Authors:** Jacek Baj, Grzegorz Teresiński, Beata Kowalska, Tomasz Krajka, Grzegorz Buszewicz, Alicja Forma, Wojciech Flieger, Kaja Hanna Karakuła, Paweł Kędzierawski, Tomasz Cywka, Jolanta Flieger

**Affiliations:** 1Chair and Department of Anatomy, Medical University of Lublin, Jaczewskiego 4, 20-090 Lublin, Poland; 2Chair and Department of Forensic Medicine, Medical University of Lublin, 20-090 Lublin, Poland; grzegorz.teresinski@umlub.pl (G.T.); g.buszewicz@umlub.pl (G.B.); pawelkedzierawski1@gmail.com (P.K.); tomaszcywka@umlub.pl (T.C.); 3Department of Water Supply and Wastewater Disposal, Lublin University of Technology, Nadbystrzycka 40B, 20-618 Lublin, Poland; b.kowalska@pollub.pl; 4Department of Production Computerisation and Robotisation, Mechanical Engineering Faculty, Lublin University of Technology, Nadbystrzycka 36, 20-618 Lublin, Poland; t.krajka@pollub.pl; 5Chair and I Department of Psychiatry, Psychotherapy, and Early Intervention, Medical University of Lublin, 20-439 Lublin, Poland; kaja.karakula@gmail.com; 6Department of Analytical Chemistry, Medical University of Lublin, Chodźki 4A, 20-093 Lublin, Poland

**Keywords:** multi-elemental analysis, trace elements, human brain, dura mater, arachnoid mater, ICP-MS

## Abstract

Metals perform many important physiological functions in the human body. The distribution of elements in different tissues is not uniform. Moreover, some structures can be the site of an accumulation of essential or toxic metals, leading to multi-directional intracellular damage. In the nervous system, these disorders are especially dangerous. Metals dyshomeostasis has been linked to a variety of neurological disorders which end up leading to permanent injuries. The multi-elemental composition of the human brain is still the subject of numerous investigations and debates. In this study, for the first time, the meninges, i.e., the dura mater and the arachnoid, were examined for their elemental composition by means of inductively coupled plasma mass spectrometry (ICP-MS). Tissue samples were collected post mortem from those who died suddenly as a result of suicide (*n* = 20) or as a result of injuries after an accident (*n* = 20). The interactions between 51 elements in both groups showed mainly weak positive correlations, which dominated the arachnoid mater compared to the dura mater. The study showed differences in the distribution of some elements within the meninges in the studied groups. The significant differences concerned mainly metals from the lanthanide family (Ln), macroelements (Na, K, Ca, Mg), a few micronutrients (Co), and toxic cadmium (Cd). The performed evaluation of the elemental distribution in the human meninges sheds new light on the trace metals metabolism in the central nervous system, although we do not yet fully understand the role of the human meninges.

## 1. Introduction

Biometals, as components of metalloproteins or metalloenzymes, participate in many metabolic processes. That is why they constitute the critical points when it comes to human health. Besides essential trace elements, the human body also accumulates toxic metals such as aluminum (Al), arsenic (As), mercury (Hg), lead (Pb), beryllium (Be), cadmium (Cd), etc. The levels of metals in the human body are subject to dynamic changes due to environmental exposure, diet, age, and the burden of disease. The distribution of trace elements varies in different types of human tissue [1,2,3,4]. Despite the existence of compensation systems, the bioaccumulation of toxic metals takes place mainly due to a reduced ability for their excretion.

It was found that inappropriate distribution of trace elements, as well as the accumulation of toxic elements in various structures of the human brain is associated with the occurrence of neurodegenerative diseases [5,6,7,8]. The relationship between the levels of such elements as copper (Cu), zinc (Zn), and iron (Fe), and the pathological processes of β-amyloid formation in Alzheimer’s disease has been proven [7]. Multi-elemental analyses of biological samples taken from patients suffering from various chronic diseases, including Alzheimer’s disease [9,10], or neurological developmental disorders (NDD) such as autism spectrum disorder (ASD) [11,12,13], schizophrenia (SZ), attention deficit hyperactivity disorder (ADHD), epilepsy, as well as from drug or alcohol abusers [14,15,16,17,18,19,20,21,22,23,24,25], have been published in specialized scientific journals. It has been proven that Zn and Fe are necessary for brain functioning, play a pivotal role in neurodevelopment, and mediate cognitive development [26,27,28]. Zn deficiency, besides Cu overload, has been identified as a risk factor for autism spectrum disorders (ASD) [29,30,31]. In turn, Cu deficiency in the brain causes Menkes disease, which is associated with demyelination and neurodegeneration [32,33].

So far, multi-element analyses have been performed using various human tissue samples [34,35,36,37,38,39]. Human brain samples were usually collected post mortem (from autopsy) from the different regions [19,20,21,40,41,42]. It was shown that the deficiency of basic trace elements, such as Zn affects the levels of other trace elements. In turn, deficiencies of Co, Cu, potassium (K), Fe, magnesium (Mg), manganese (Mn), molybdenum (Mo), and selenium (Se), can increase the absorption of toxic metals. The pairwise correlations of elements of varying intensity have been described in the literature in relation to different tissues and body fluids [43,44,45].

Analyses of the multi-element composition of the meninges have not been reported so far. To fill this knowledge gap, we investigated, for the first time, whether significant differences in trace metal levels are visible in the substructures of the meninges (the dura mater and the arachnoid mater). The meninges, as the tissue surrounding the brain, may reflect long-term exposure to fluctuations in the composition of micronutrients, macronutrients, toxic elements, or their bioaccumulation (similar to hair, skin for example). The aim of the study was to test this hypothesis. The test group consisted of subjects who have not been burdened with serious diseases, i.e., the victims of road accidents (control group, *n* = 20) and suicides (case group, *n* = 20). All of the autopsies with cases of injuries located within the head were excluded from both groups. The element differentiating both groups was the occurrence of a deep nervous breakdown leading to suicide.

According to the World Health Organization (WHO) estimates, suicide is one of the leading causes of death in the world. In 2019, 700,000 people committed suicide, which is more than those that died from HIV or malaria. Overall, it has been estimated that one in 100 deaths is due to suicide. Worryingly, suicide was the fourth leading cause of death among young people aged 15–29. Both mental and neurological disorders have many causes. The one-dimensional approach to the etiology of these diseases has fewer and fewer supporters. Most studies show that there is an association between suicidal behavior and the deregulation of the hypothalamic-pituitary-adrenal (HPA) axis [46]. The reasons for suicidal behavior are also ascribed to the reactivity of cortisol involved with chronic stress and a reduced number of glucocorticoid receptors (NR3C1). However, even proven genetic predisposition does not exclude the influence of the environment on mental health, especially exposure to environmental toxins [47]. Many authors confirm a strong correlation between urbanization and the occurrence of psychopathology, which is usually diagnosed in adolescence [48,49].

As the meninges’ elemental composition may indicate long-term disturbances in the electrolyte balance in the brain area, the obtained results allow us to answer the following questions: What kind of differences exist in the dura and arachnoid elemental composition? What kind of disturbances in the elemental composition occur in subjects experiencing a severe nervous breakdown that leads to suicide in comparison to controls? The performed ICP-MS analysis of the meninges included 51 elements. The statistical analysis of experimental results was conducted to identify bioaccumulating elements and inter-elemental interactions.

## 2. Results

### 2.1. Correlations of Average Trace Element Concentrations in the Meninges for the Whole Population of Subjects (n = 40)

In independently collected and independently quantified samples of each brain meninx (in two independent replications), no statistically significant differences in the elemental composition were found. Tests showed sporadic differences in the arachnoid mater for the level of Hg at a significance level of *p* < 0.05. This result indicates a rather uniform distribution of elements in every brain meninx. Noteworthy is the high standard deviation of the measurements and the lack of a normal distribution in the studied groups, which proves a high variability varying from 0.05 [ppb] for thulium (Tm) to 10,180 [ppb] for Al, 37,020 [ppb] for iron (Fe) and 978 [ppm] for calcium (Ca). Among the 51 examined elements, two, ie arsenic (As) and palladium (Pd), were below the detection limit and were not taken in further statistical studies.

The elemental composition of all subjects was independently correlated for the dura mater (T) (Figure 1a), the right arachnoid (PP) (Figure 1b), and the left arachnoid mater (PL) (Figure 1c). The correlations presented in the three graphs show that some elements are strongly correlated with each other, especially in the arachnoid meninx on both the left and right sides (Figure 1a,b). The correlation obtained for a dura mater is clearly different. The correlations are much less strong and there are almost no negative correlations.

The correlation matrices for the left and right arachnoid maters are very similar to each other. Within the arachnoid mater, positive correlations of greater importance concern mainly Be, Al, vanadium (V), gallium (Ga), Se, and barium (Ba), which correlate well with rare earth metals. Additionally, positive correlations are seen for the following pairs: rubidium (Rb)-K, sodium (Na)-K, Mg-phosphorous (P), Be-V. The strength of these correlations weakens in the case of the dura mater; however, additional positive pairwise correlations are formed here, regarding macronutrients such as Mg-Ca, P-Ca, and correlations of the toxic Pb with elements such as Mg, P, Ca, Zn, Ba.

It should be emphasized that the positive correlations observed in the dura mater in relation to the Mg-P, Mg-Ca, P-Ca pairs, and in the arachnoid meninx in relation to the following pairs Rb-K, Na-K, Mg-P, Be-V are mainly due to the chemical similarity of the paired elements.

### 2.2. Statistically Significant Differences between Meninges

The finding of the differences in the elemental composition of the arachnoid and dura between the two groups has been the subject of the statistical investigation. In the interpretation of the statistical tests, a threshold of *p* = 0.05 was used to answer the question of whether the observed differences between the groups were statistically significant (*p* < 0.05) or whether the differences were a matter of chance (*p* > 0.05).

Table 1 includes the results of three tests: the Ansari-Bradley test (ANS), the Wilcox Mann-Whitney rank-sum test (WILCOX), and the Brunner-Munzel test (BMP), along with the parameters of the statistical analysis. The WILCOX test investigates the distribution of characteristics of the studied groups under the H_0_ hypothesis that the comparing distribution functions are equal. Assumptions required for this test include the statistical independence of observations, the ordinal scale of measure, and the statistical equality of scale parameters, which is usually checked by ANS test. On the other hand, the BNP test assumes that the H_0_ that is observed in one of the groups are higher than those in the second. Thus, the ANS test p-values higher than 0.05 indicate that the application of the WILCOX test is correct, whereas the WILCOX and BMP tests *p*-values lower than 0.05 make sure of a rejection of the H_0_ hypothesis. The rejection of the H_0_ hypothesis, and hence the existence of the statistically significant differences between the studied groups, was detected only for a few elements belonging to the alkali metals Na, K, cesium (Cs), Rb, the transition metals Co and Cd, metalloid antimony (Sb), and lanthanides praseodymium (Pr), samarium (Sm), dysprosium (Dy), terbium (Tb), and erbium (Er). The level of alkali metals (Na, K, Rb, Cs) was significantly lower in the meninges of suicides compared to those who died suddenly in an accident. This may indicate serious electrolyte disturbances at the macronutrient level in the case group.

### 2.3. Statistically Significant Differences between Elemental Composition of Meninges for Controls and Cases

Owing to the packages available in R, it is possible to perform the principal component analysis (PCA) and their visualization. Graphical PCA analysis enables the representation of an n-dimensional data set in a two-dimensional space. PCA analysis was performed for the ICP-MS data obtained for the dura mater and arachnoid mater (Figure 2). Although the PCA graph relating to the first two components explains about 40% of the variance, it allows for the separation of the imposed categories (suicide/death in an accident). As can be seen in Figure 2, in the case of the dura mater, the category fields overlap almost completely, and in the case of the arachnoid meninges, they are clearly separated.

In order to identify differences in the elemental composition of individual brain meninges, pairwise correlations were examined in both examined groups. The results that were statistically significant are presented in Table 2. It should be noted that no statistically significant differences were found in the PP and PL composition in both study groups. The differences concerned Na and bismuth (Bi), the levels of which differed significantly between T and PL in the suicide group, and Na, Mn, and Co, which were significantly different in the T and PR of this group. Many more differences can be distinguished in the group of those who died suddenly from accidents. In this group, there were differences between T and PL which were observed for Mg, tin (Sn), lanthanum (La), thallium (Tl), and between T and PR, which were significant for Mg, Cd, Ba, La, Pr, neodymium (Nd), Sm, Ga, and Tb.

Based on the collected data, it can be noticed that a clear difference between the studied groups appears in the case of rare earth elements. In the case of suicide death, the levels of these elements increase almost twofold, and the differences in the elemental composition between the arachnoid and the dura mater are no longer statistically significant.

In our study, a statistical difference was found in the content of lanthanides (Tb, Dy, Er, Sm, Pr) in the meninges between the suicide and the naturally deceased groups. In the group that died by suicide, a significant increase in the content of lanthanides in the subarachnoid dura was noted, while their content in the outermost dura mater closest to the skull bone is significantly lower compared to those who died due to accidents. This first study allows us to conclude that there is a significant affinity of selected elements from the lanthanide family to the arachnoid meninx. This affinity increases in the group of those who died by suicide, where the differences in the levels of these elements between the dura mater and arachnoid surprisingly disappear.

### 2.4. Statistically Significant Differences between Elements’ Ratios for Suicide Cases in Comparison to Sudden Death from Accidents

Cd neurotoxicity affects the peripheral function (PNS) [50], and the CNS [51], and leads to disorders that are manifested by neurological and psychological clinic symptoms. There is a hypothesis that Cd contributes to the dysfunction of the blood-brain barrier (BBB), which is responsible for the effects of Cd on the CNS. Observed bioaccumulation of Cd in the arachnoid meninx, which is in high contact with the circulation of the cerebrospinal fluid, may indicate a protective role of the meninx for the brain against neurotoxic metal. The absence of this sorption through the meninges would not be beneficial because this toxic metal xenobiotic could threaten the inner structures of the brain.

The study examined the ratios of Cd to chosen micro-elements taking part in the Cd metabolism (Table 3). Among the metals that play a protective role against Cd, the most often mentioned are those with similar physicochemical properties, such as zinc (Zn) and selenium (Se). The mechanism of the protective action of Zn consists in the induction of metallothionein III synthesis, which plays a detoxifying role. Se, in turn, alleviates the oxidative stress caused by Cd as a cofactor of the enzyme glutathione peroxidase (GPx) [52,53,54]. As mentioned earlier, the level of Cd in the arachnoid differs statistically significantly between cases and controls (*p* = 0.016; *p* = 0.043). As can be seen (Figure 3), Zn, Se, as well as Cu, do not change the statistically significant difference existing between studied groups (*p* (Cd/Zn) = 0.017; *p* (Cd/Cu) = 0.016; *p* (Cd/Se) = 0.043). In turn, Ca, Fe, and P eliminate these differences (p values for the ratios: Cd/Ca; Cd/Fe; Cd/P are higher than 0.05). This observation partially confirms previous reports. The effect of Cd on Ca metabolism is already known. Cd can pass into neurons via voltage-gated calcium channels [55], regulate Ca^2+^ signaling by inhibiting the activity of 1,4,5-triphosphate (IP3) and ryanodine receptors [56], and promote the efflux of calcium from the sarcoplasmic reticulum. Thus, the cooperation between these elements (Cd; Ca) has been confirmed in our study.

## 3. Discussion

It is known that heavy metals such as Cd, Hg, Cr, Pb are commonly found in the environment. The urbanized areas, as well as agriculture, are the main source of high heavy metal levels [57,58]. These metals also are found in clothes as well as facial masks.

Recently, there was a relevant article regarding an ICP-MS analysis of surgical and face masks. This is relevant as, during the current pandemic, face masks were commonly used in order to protect against COVID-19. The study evaluated the possibility of the transfer of trace elements into the human body based on saliva leaching and breathing experiments [59]. Most frequently, exposure to Cd, Pb, and other heavy metals poisoning are pointed out as causes of mental dysfunctions [60].

Undoubtedly, heavy metals like Cd induce structural changes in the brain like a diminution in the total cortex volume, white matter [61], magnification arrangement of the cerebral ventricles, changes in gray and white matter, and incorrect laminar organization [60]. Cd neurotoxicity has been confirmed to lead to neuropsychiatric disorders. A statistically significant increase in the level of Cd in the blood was found in the manic state of patients in the depressive phase and in the bipolar group [62]. An unclear relationship between Cd and homocysteine has also been described [63,64]. Cd bioaccumulation is the result of exposure to this element. After being absorbed through the lungs or intestines epithelium, Cd enters the systemic circulation. It is known that high doses cause cell damage leading to cell death; however, at low doses, Cd can only modulate certain processes [65,66]. The first study that found a relationship between the level of blood Cd (BCd) and the onset of depressive symptoms dates back to 2014 [67]. It should be emphasized that the measurement of blood Cd levels only indicates actual exposure as the blood Cd half-life is short, some three to four months [68]. However, Cd can accumulate in tissues for a long time due to its long half-life (30 years). Cross-sectional studies covering approximately 3,000 adults showed that people from the highest BCd quartile had a higher probability of developing depression symptoms (odds ratio 2.79, 95%; confidence interval 1.84–4.25) than people from the lowest BCd quartile. Most of the published reports describe the presence of elevated levels of toxic metals in tissues (bones) and body fluids (blood) in patients [61,62,69,70,71,72,73,74,75,76,77,78,79,80] suffering from depression, schizophrenia, anxiety, phobias, etc. The human brain is particularly sensitive to environmental toxins. This is due to the fast-metabolic rate in the brain, the high content of fat and sulfur amino acids complexing heavy metals [81].

In the group of suicides, significantly different levels of Cd were observed compared to the control group. In our study, no elevated Cd levels were found in the suicide group. The level in this group was even significantly lower compared to those who died in accidents. However, this does not indicate the level of safety in relation to Cd in both groups, as there are no reference values. Due to its long Cd half-life and the confirmed contribution of this element in the development of depression, future studies of cerebrospinal fluid, blood, and other regions of the brain should be expanded.

The performed study showed changes in the distribution of some elements within the meninges in the group of suicides compared to the control group. Most of the observed correlations were positive and clearly stronger in the arachnoid meninx. This may suggest that the transport within the dura mater and the possible accumulation of elements occur here to a lesser extent compared to the arachnoid meninx. Moreover, the above correlation matrices clearly show the different roles of the two meninges when it comes to the storage of biometals. The dura mater appears to have greater sequestration and buffering capacity for toxic metals (Pb). This phenomenon can occur with intrinsically toxic or excess metals to defend the host cells. Due to accumulation, toxicity can be avoided. This is especially important in case of the burden by highly toxic metals such as Al, Cd, and Pb because of their proven adverse influence on cognition [82]. Such an effect is possible not only due to compartmentalization, but also the involvement of specific molecular systems for metal storage. The components of the system are mainly metallothioneins and small detoxification molecules containing cysteine [83,84]. It turns out that the Pb content in the dura mater positively correlates with metals such as Mg, P, Ca, Zn, and Ba. This is understandable, given that the mechanism of Pb toxicity is mediated exactly by Zn, Mg, or Ca, which can be replaced with Pb. This type of relationship has been previously confirmed, for example, in children with ASD [85]. Our study clearly shows similar pairwise correlations. The storage of biometals with signaling functions (e.g., Ca) also seems understandable from the point of view of essential metals homeostasis.

While the bioaccumulation of toxic metals, such as Cd, confirms the existing research, disturbances in the distribution of metals from the lanthanide family are a new observation. The lanthanides family includes elements such as La, Ce, Pr, Nd, promethium (Pm), Sm, europium (Eu), Gd, Tb, Dy, holmium (Ho), Er, thulium (Tm), ytterbium (Yb), and lutetium (Lu). These metals have similar physical and chemical properties. Their ions are in the +3 oxidation state, and rarely in the +2 and +4 oxidation states. Besides, they have many unpaired electrons, which are a kind of small magnet with meaning magnetic moments which are several times greater than that of transition metals. Lanthanides have many applications. Besides the production of glasses and catalysts, they are used as batteries, magnets (e.g., strong neodymium-based magnets), electricity-generating devices, and phosphors. Due to their luminescent properties, their applications also include optoelectronic materials like lasers (the best known is the Nd: YAG laser), optical amplifiers or TV picture tubes. It is worth noting that the lanthanides (excluding radioactive Pm) have low toxicity and can be used in medicine for diagnostics and therapy. In the past, the use of lanthanides in diagnostics as contrast agents, in therapy for the treatment of burn wounds (Flammacerium), neoplasms, hyperphosphatemia (Fosrenol), immunity disorders, cancer (Gd-Motexafin), and osteoporosis was emphasized [86]. For instance, Gd^3+^ ions are used in magnetic resonance imaging (MRI) as a contrast agent [87,88], and La^3+^ and Ce^3+^ compounds are tested as potential anti-cancer agents [89,90].

So far, there is no data available on the determination of lanthanides in the human brain. There is, therefore, a need to study various brain structures for lanthanides to solve the problem of their distribution and their role in brain function. Without doubt, the lanthanides have received little attention so far when it comes to analyzing the elemental composition of human tissues. In our previous reports, we have shown their presence in fluid collected from the anterior chamber of patients undergoing cataract surgery [38,91,92]. This fluid, which is partly a plasma filtrate, had a lanthanide content that was two or three orders of magnitude lower than in the meninges. This comparison confirms the specific affinity of the lanthanides to the arachnoid mater. The lanthanides’ role is still unclear.

In 1981, El-Fakahany [93], in experiments carried out on murine neuroblastoma cells (clone N1E-115), showed that lanthanides (Tb, Eu, Nd, La) increase the efficacy of muscarinic acetylcholine receptor agonists by interacting with Ca^2+^ binding sites. It should be emphasized that receptors of this type are mainly located in the central nervous system. In turn, it has been shown that free lanthanides (Tb, Gd, Eu) are able to activate myosin, troponin: tropomyosin, while simultaneously they do not compete with calcium in muscle fibers [94,95]. As can be seen, the hypothesis that lanthanides act as calcium analogs in biological systems requires more studies. To our knowledge, their role in psychiatry has not been examined by anyone so far.

The bioaccumulation of metals from the lanthanide family may result from increased exposure to these metals or disturbances in their transport through biological membranes. It is known that the exchange of high-charge molecules via BBB between blood and brain interstitial fluid, the choroid plexus epithelium, and the arachnoid epithelium is not very fast, and usually it significantly slowed down because of their hydration. Moreover, ion homeostasis in the brain is mediated by protein transporters and endothelial cells. Elucidation of the molecular mechanisms of metal ions’ influence on the permeability of BBB requires further research on larger groups of patients from different environments and the extending of the scope of research to tissues from other regions of the brain. The probable explanation we can suggest is the feature of heavy metal ions to intracellular ROS generation and the induction of oxidative stress causing the increase in the permeability of the cell membrane.

Based on the results of our study, meninges can potentially have an additional role in human organisms apart from being a protective barrier for the CNS. Analogically, quite recently, mesentery was announced as a ‘new organ’ and this observation was based on both—the anatomical knowledge and clinical practice that confirmed that, in fact, the mesentery is a continuous collection of tissues with quite new functions that previously were not noticed [96]. This is similar to the case of chiasma opticum, which contains a higher concentration of Ca than the other areas of the white matter. The increased Ca levels for the choroid plexus and pineal gland were reported by François Foulquier et al. [84]. There are different elemental compositions of the lens in comparison to the fluid of the anterior chamber of the eyes [97]. Some heavy metals such as Pb, Hg, and Cd are selectively accumulated in the hair of autistic children [85,98]. V can selectively replace P in hydroxyapatite in bones, [99]. Hemoglobin exhibits increased affinity to Fe. Thus, the tissue content does not always reflect the composition of surrounding biofluids. In our study, we also assumed that meninges could potentially have other functions in addition to the protective ones, and similarly to some tissues and organs are able to selectively accumulated metals. However, our hypothesis in the context of the meninges should be further evaluated, taking other variables into account.

## 4. Materials and Methods

### 4.1. Population

The brain tissues analyzed for the metal contents were collected at the Department of Forensic Medicine, University of Lublin, Lublin, Poland. The samples were frozen deeply (in liquid nitrogen) to avoid any metabolism. The study was approved by the Local Ethical Committee Medical University of Lublin, Poland, approval no. KE-0254/152/2021, and the tissue collection was approved by the prosecutor’s office. The research has been carried out in accordance with The Code of Ethics of the World Medical Association, Declaration of Helsinki for experiments involving humans. Meningeal samples were taken post mortem during necropsy from patients who died suddenly as a result of injuries sustained during an accident (*n* = 20) or via suicide by hanging (*n* = 20). The control group included the cases of sudden death victims who died due to road accidents including deaths due to injuries of the thorax (most commonly damage/rupture of the aorta due to penetrating/blunt force and penetrating injuries of the thoracic viscera) and abdomen (blunt/ penetrating abdominal trauma). All of the autopsies with cases of injuries located within the head and/or neck and/or spinal group were excluded from both groups. The study group included the cases of death due to suicide by hanging without any injuries within the head, neck, or vertebral column (Figure 4). The demographic characteristics of the patients’ groups is collected in Table 4.

### 4.2. Sample Collection

Dura mater and arachnoid mater were collected from the right and left frontoparietal parts of the brain. The arachnoid matter was collected from the right and left hemisphere separately, while the dura mater was collected as a strand which was located over the right and left hemispheres near the border between the frontal and parietal lobes. We used ceramic scissors to make a parasagittal cut through the dura mater about 2 cm lateral to the midline. First, only cut the dura mater was cut, and the arachnoid mater remained intact. This cut should be lateral and parallel to the lateral edge of the superior sagittal sinus. We extended the cut to the frontal bone anteriorly and the transverse sinus posteriorly. Then we duplicated the parasagittal cut on the opposite side of the cadaver. The next step was the cutting lateral and parallel to the lateral edge of the superior sagittal sinus, and we duplicated the parasagittal cut on the opposite side of the cadaver. The result was a median strip of dura mater containing the superior sagittal sinus and four flaps of dura mater that are similar to the scalp flaps. The arachnoid mater loosely covers the brain and spans across the fissures and sulci. We used ceramic scissors to make a small cut through the arachnoid mater over the lateral surface of the brain. Then we elevated the arachnoid mater with the ceramic tool and collected the sample.

### 4.3. Sample Preparation

Wet mineralization of each 0.3–0.5 g sample (cut with a ceramic knife) was performed via the addition of 7 mL of 69% suprapur nitric acid HNO_3_ (Baker, Sanford, ME, USA), followed by heating to 190 °C in close Teflon containers using the microwave mineralization system TOPEX (PreeKem, Shanghai, China). After mineralization, 1 mL of HCl (Merck, Darmstadt, Germany) was added to stabilize some elements (As, Hg, Se, Mo, Tl, Ag). Finally, the samples were diluted to 25 mL by ultrapure water obtained in the Milli-Q (Millipore, Darmstadt, Germany) purification system.

### 4.4. ICP-MS Measurements

The inductively coupled plasma mass spectrometer Agilent 8900 ICP-MS Triple Quad (Agilent, Santa Clara, CA, USA) was employed for elemental analysis. Most of the elements were analyzed in He mode (5.5 mL/min helium flow), although Se and As were analyzed in O_2_ mode (gas O_2_ flow rate-30%). The plasma was working in general-purpose mode with 1.550 kW RF power, the nebulizer gas flow was 1.07 L/min, the auxiliary gas flow was 0.9 L/min, and the Plasma gas flow was 15 L/min. Acquisition time was from 0.1 to 2 s depending on the predicted concentration of the element.

Due to the lack of certified reference material, the internal standard ISTD (Sc, Y, Lu) with a concentration of 0.5 ppm was used for the analysis. ISTD was added automatically using a standard mixing connector, the so-called mixing tee. The obtained recoveries were in the range of 80–120%. ICP commercial analytical standards were purchased from Agilent Technologies, Santa Clara, CA, USA (Multi-Element Calibration Standard 2A-Hg, Environmental Calibration Standard, Multi-Element Calibration Standard 2A), Merck Millipore, Darmstadt, Germany (ICP-Multi-Element Calibration Standard XVII, ICP-Multi-Element Calibration Standard VI, Phosphorus ICP standard), Honeywell Fluka™( Charlotte, North Carolina, USA) analytical standards (Platinum Standard for ICP, Palladium Standard for ICP), and Inorganic Ventures, Christiansburg, VA, USA (Rare Earth, Standards).

The validation protocol of this analytical method was presented as Appendix A. The report includes validation parameters (background equivalent concentration-BEC, detection limit-DL, internal standard-ISTD, calibration equation with correlation coefficient-R) and individual curves for each examined element, which are the basis for their quantification.

### 4.5. Statistics

All statistical analyses were made in R language on the R-Studio platform. For testing normality, we used the Shapiro-Wilks test, the Anderson-Darling test, and the Lilliefors test. Because all of the observations were not present under normal distribution, we used non-parametric tests only. In order to compare two or more statistically independent observations, we used the Wilcox Mann-Whitney rank-sum test and the Brunner-Munzel test, whereas for testing statistically dependent observations, the Friedman test was used. For the correctness of the Wilcox Mann-Whitney test (assumption on equality of parameters of scale), the Ansari-Bradley test was performed.

## 5. Conclusions

We measured the levels of 51 elements in meningeal samples taken at autopsy from patients who died suddenly from accidents or as a result of suicide. Brain samples were prepared in the same way and analyzed using the same technology. They can therefore be useful for comparative analysis.

Our results confirm significant differences in trace metal binding by the arachnoid mater compared to the dura mater. Correlation matrices made for both studied groups together are characterized by few and quite weak positive correlations and no significant negative correlations. In the case of the dura mater, which is located mostly externally, just below the skull bones, the correlations are insignificant. However, for the arachnoid mater, the correlations are much stronger and occur mainly between macronutrients and rare earth elements. This may suggest that the transport within the dura mater and the possible accumulation of elements occur here to a lesser extent compared to the arachnoid meninx.

The differences between cases and controls concern mainly the alkali metals Na, K, Cs, Rb, and other metals like Co and Cd, Sb, levels of which were significantly lower in those who died by suicide, and the lanthanides Pr, Sm, Tb, Dy, and Er, whose levels were statistically significantly higher in this group. Bioaccumulation of metals from the lanthanide family as well as toxic Al and Cd may result from increased exposure to these metals or disturbances in their transport through biological membranes. Since the disturbing metals’ homeostasis is related to diverse neuropathologies and behavioral dysfunctions, our observations with statistical data treatment could be the starting point for further studies by physicians, especially psychiatrists.

## Figures and Tables

**Figure 1 molecules-27-01911-f001:**
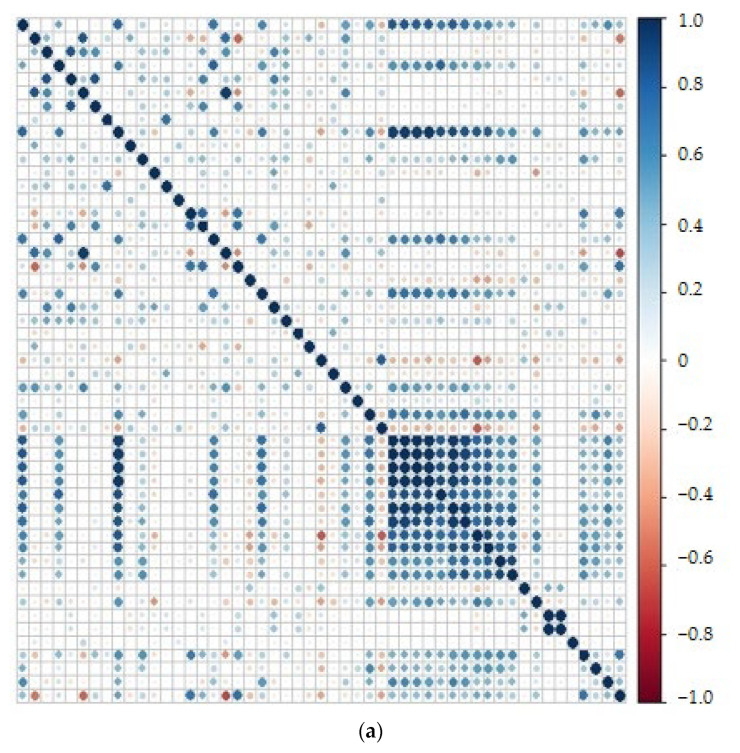
Spearman rank-order correlation matrices of average trace elements in the following order-Be, Na, Mg, Al, P, K, Ca, Ti, V, Cr, Mn, Fe, Co, Ni, Cu, Zn, Ga, Rb, Sr, Zr, Se, 78–94 Se, Mo, Ag, Co, Sn, Sb, Cs, Ba, La, Ce, Pr, Nd, Sm, Eu, Gd, Tb, Dy, Ho, Er, Tm, Yb, Hf, Pt, Hg_201_, Hg_202_, Tl, Pb, Bi, Th, U, quantified in the meninges: the left arachnoid (**a**), and the right arachnoid (**b**) mater, and the dura mater (**c**) of all subjects (*n* = 40).

**Figure 2 molecules-27-01911-f002:**
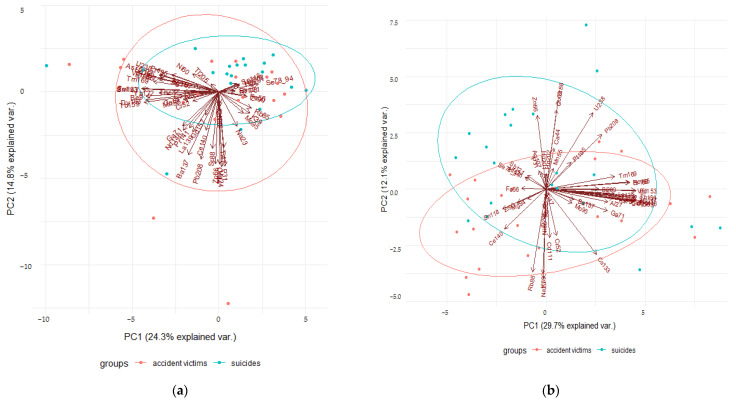
Principal component analysis (PCA) biplots of ICP-MS data from the dura matter (**a**), and the arachnoid mater (**b**). Only the first two principal components (PC1, PC2) are shown with their respective variation in percentage.

**Figure 3 molecules-27-01911-f003:**
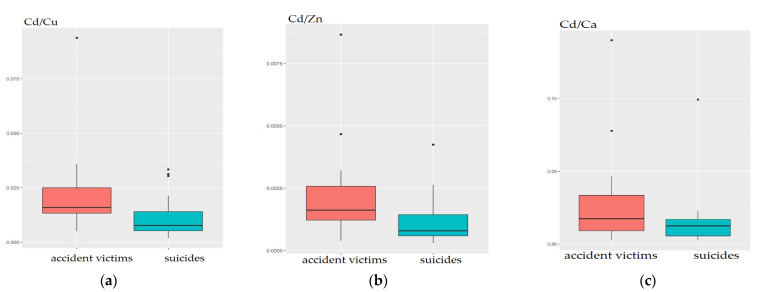
Statistically significant differences in Cd/Cu (**a**), Cd/Zn (**b**), Cd/Ca (**c**) ratios in the arachnoid of suicides (cases) in comparison to the accident victims (controls).

**Figure 4 molecules-27-01911-f004:**
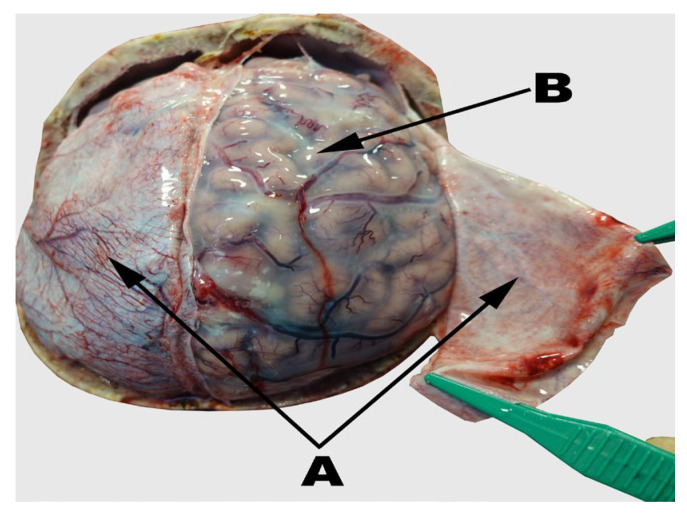
The meninges’ picture done during autopsy: the dura mater (**A**), and the arachnoid (**B**) mater.

**Table 1 molecules-27-01911-t001:** The results of the statistically significant differences between controls (*n* = 20) and cases (*n* = 20) considering selected elementals performed by Ansari-Bradley test *p*-value (ANS), Wilcoxon Mann-Whitney test *p*-value (WILCOX), and Brunner-Munzel test *p*-value.

Element	Tissue	Mean Value for Cases	Mean Value for Controls	ANSp	WILCOXp	BMp
Na ^1^	T	661 (252)	846 (224)	0.984	0.044	0.034
	PL	369 (235)	710 (310)	0.980	0.000	0.000
	PP	415 (269)	659 (254)	0.795	0.006	0.003
K ^1^	T	581 (252)	741 (226)	0.766	0.047	0.040
	PL	507 (302)	739 (231)	0.784	0.005	0.003
	PP	541 (315)	710 (194)	0.283	0.034	0.029
Rb ^1^	PL	414 (236)	571 (179)	0.746	0.011	0.007
Cs ^2^	T	1.35 (0.68)	1.73 (0.56)	0.682	0.047	0.039
	PL	1.27 (0.72)	1.74 (0.76)	0.901	0.044	0.037
Co ^2^	T	5.12 (1.83)	8.70 (4.61)	0.449	0.003	0.001
	PL	5.94 (2.48)	10.46 (12.39)	0.469	0.055	0.047
Cd ^2^	PL	7.54 (4.19)	11.09 (4.72)	0.566	0.016	0.010
	PP	7.55 (5.76)	11.26 (6.49)	0.469	0.043	0.046
Sb ^2^	PP	0.55 (2.40)	0.55 (1.24)	0.008	0.036	0.035
Pr ^2^	PP	0.84 (0.95)	0.45 (0.78)	0.769	0.054	0.044
	T	1.15 (1.98)	4.53 (13.22)	0.666	0.037	0.033
Sm ^2^	PP	0.67 (0.71)	0.35 (0.69)	0.792	0.056	0.049
Er ^2^	PP	0.81 (0.63)	0.50 (0.42)	0.770	0.054	0.044
Dy ^2^	T	0.27 (0.52)	0.51 (0.53)	0.562	0.038	0.033
Tb ^2^	T	0.05 (0.09)	0.1 (0.1)	0.475	0.050	0.049

Abbreviations: T—the dura mater, the arachnoid mater, PP—the right arachnoid, PL—the left arachnoid mater; ^1^ ppm-µg/g, ^2^ ppb-ng/g.

**Table 2 molecules-27-01911-t002:** Results of statistical differences (*p*-values) in elemental composition between meninges for cases (*n* = 20) and controls (*n* = 20) enrolled in the study. Abbreviations: PR-the right side of the arachnoid meninx, PL-the left side of the arachnoid meninx, T-the dura mater.

Element	Sudden Death Due to Suicide				Sudden Death from Accidents			
	Friedp	PR/PL	T/PL	T/PR	Friedp	PR/PL	T/PL	T/PR
Na	0.002	0.686	0.008	0.008	0.143	0.996	0.219	0.054
Mg	0.066	1.000	0.062	0.071	0.003	1.000	0.000	0.001
Ca	0.009	0.325	0.027	0.296	0.018	1.000	0.042	0.077
Mn	0.024	1.000	0.036	0.042	0.229	1.000	0.032	0.148
Fe	0.000	1.000	0.001	0.000	0.000	1.000	0.004	0.009
Cu	0.015	1.000	0.081	0.002	0.368	1.000	1.000	1.000
Mo	0.000	0.589	0.000	0.000	0.000	1.000	0.000	0.000
Cd	0.000	1.000	0.000	0.000	0.000	1.000	0.000	0.000
Sn	0.206	0.317	0.424	1.000	0.126	1.000	0.043	0.205
Ba	0.946	1.000	1.000	1.000	0.076	1.000	0.506	0.010
La	0.211	1.000	0.130	0.686	0.003	1.000	0.048	0.019
Pr	0.607	1.000	0.502	1.000	0.007	1.000	0.148	0.014
Nd	0.946	1.000	1.000	1.000	0.015	1.000	0.490	0.039
Sm	0.607	1.000	1.000	1.000	0.006	1.000	0.950	0.035
Eu	0.796	1.000	0.544	1.000	0.037	1.000	1.000	0.061
Gd	0.958	1.000	1.000	1.000	0.024	0.773	0.829	0.035
Tb	0.573	1.000	1.000	1.000	0.054	0.826	0.888	0.028
Dy	0.678	1.000	1.000	1.000	0.015	1.000	0.620	0.079
Hg	0.047	1.000	0.047	0.088	0.150	1.000	0.025	0.312
Tl	0.092	0.389	0.071	1.000	0.065	0.219	0.028	1.000
Bi	0.009	0.244	0.027	0.851	0.241	0.164	0.370	1.000
U	0.000	0.851	0.002	0.003	0.003	1.000	0.001	0.002

**Table 3 molecules-27-01911-t003:** Differences between controls and cases considering selected elemental ratios. The analysis was performed by the Wilcoxon Mann-Whitney rank sum test (x-value of the test, p-value, Ansari-Bradley test gives all values greater than 0.05); IQR means interquartile range.

Element Ratio	Wilcox x	Wilcox *p*	Median Cases	IQR Cases	Median Controls	IQR Controls
Cd/Zn	93	0.017	0.001	0.001	0.002	0.001
Cd/Se	49	0.043	5.345	11.419	12.346	35.369
Cd/Cu	92	0.016	0.008	0.009	0.016	0.012
Cd/Ca	125	0.169	0.013	0.011	0.018	0.024
Cd/P	124	0.159	0.011	0.011	0.018	0.013
Cd/Fe	116	0.098	0.000	0.000	0.000	0.000

**Table 4 molecules-27-01911-t004:** Demographic characteristic of the patients’ groups enrolled in the study.

Group	Gender	*n*	%	Min–Max Age	Median Age	Mean Age ± SD
cases (*n* = 20)	Female	2	10	13–17	15	15 ± 2.83
	Male	18	90	20–62	49	44.67 ± 13.58
controls (*n* = 20)	Female	3	15	29–83	58	56.67 ± 27.02
	Male	17	85	18–93	57	55.46 ± 16.67

## Data Availability

The data presented in this study are available upon request from J.B.

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
