# Peer review of "ICP-MS Multi-Elemental Analysis of the Human Meninges Collected from Sudden Death Victims in South-Eastern Poland"

_molecules, 2022, doi:10.3390/molecules27061911_

Round 1

Reviewer 1 Report

The paper of Baj et al. provides some trace element data in the brain meninges from suicidal individuals vs. people deceased from accidents as controls. The paper may pose some interest for the readership. However, in the current form, the paper has some serious concerns which interfere with soundness and clarity to evaluate the paper adequately. It seems that the paper was revised already (there is some text marked red, but I have not seen the original). Anyway, I recommend major revision.

1) I may be wrong, but it seems that consent from the relatives should have been requested for such a study.

2) Data presentation does not provide a way to understand it. Figure 1 is too small to be able to see anything there. So nothing can be evaluated regarding the correlations. Figure rounding in all Tables does not make any sense. It looks that the output values were inserted in the paper just without any rounding. Remember, the rounding should correspond to the actual accuracy. Why different tests were used for p-value in Table 1? 

3) The paper is extremely wordy, there are paragraphs in the Result section which just recite some facts of the elements which has nothing to do with the results and can be just removed and not moved to the discussion. Focus on your main thing. The introduction may be shortened twice in my opinion, the initial paragraphs may be just removed completely, in my opinion. What is red later seems to be more relevant. Instead of giving some random background information, the authors should focus on properly stating their hypothesis. Currently, the reader needs to guess it.

4) Figure 2 is unclear

5) Many statements in the paper are rather obscure. Please reword those. I do not provide the examples here because in my opinion a lot of redundant sentences and paragraphs should be just removed completely.

6) The authors quote the "validation protocol" in the supplement. However, the supplement shows only a bunch of calibration curves which do not prove any trueness or accuracy of the analysis of a biomedical sample. Please provide the validation based on the recommendations of BIMP etc., e.g. certified reference materials, spike recoveries etc.

Author Response

The paper of Baj et al. provides some trace element data in the brain meninges from suicidal individuals vs. people deceased from accidents as controls. The paper may pose some interest for the readership. However, in the current form, the paper has some serious concerns which interfere with soundness and clarity to evaluate the paper adequately. It seems that the paper was revised already (there is some text marked red, but I have not seen the original). Anyway, I recommend major revision.

All authors are very grateful for the comprehensive review and insightful analysis of our work. The reviewer's suggestions have been taken into account to prepare the revised version of the manuscript. This helped us to re-think our work again.

1) I may be wrong, but it seems that consent from the relatives should have been requested for such a study.

It is not required. The decision is made by Prosecutor and the Local Ethical Committee.

2) Data presentation does not provide a way to understand it. Figure 1 is too small to be able to see anything there. So nothing can be evaluated regarding the correlations. Figure rounding in all Tables does not make any sense. It looks that the output values were inserted in the paper just without any rounding. Remember, the rounding should correspond to the actual accuracy.

Yes. We agree with this suggestion. The Figure was corrected. The values in Tables were rounding appropriate.

Why different tests were used for p-value in Table 1?

As for some elements, such as Sb, the Ansari test was less than 0.05, the Brunnel-Munzel test results are more reliable in these cases (assumptions of the Wilcoxon test are not met). In other cases, the assumptions of the more famous and popular Wilcoxon test are met.

3) The paper is extremely wordy, there are paragraphs in the Result section which just recite some facts of the elements which has nothing to do with the results and can be just removed and not moved to the discussion. Focus on your main thing. The introduction may be shortened twice in my opinion, the initial paragraphs may be just removed completely, in my opinion. What is red later seems to be more relevant. Instead of giving some random background information, the authors should focus on properly stating their hypothesis. Currently, the reader needs to guess it.

Thank you for this suggestion. We shortened the introduction part leaving the ones which are important for the background of our investigation.  We do hope that the reviewer will accept our improvements.

4) Figure 2 is unclear

The figure was deleted and PCA biplots were inserted instead. This type of chart will certainly be more understandable. Thank you for this suggestion.

5) Many statements in the paper are rather obscure. Please reword those. I do not provide the examples here because in my opinion a lot of redundant sentences and paragraphs should be just removed completely.

We reworded some chapters especially subchapter 2.4 leaving information necessary to understand our choices for subsequent investigations. 

6) The authors quote the "validation protocol" in the supplement. However, the supplement shows only a bunch of calibration curves which do not prove any trueness or accuracy of the analysis of a biomedical sample. Please provide the validation based on the recommendations of BIMP etc., e.g. certified reference materials, spike recoveries etc.

Thank you for this suggestion.  Indeed we should add some important information considering quantification. Due to the lack of proper Certified Reference Material, the traceability was assessed by a standard addition procedure for a real sample. A recovery of 80–120% was considered acceptable for all the examined elements.  In the supplementary file, we added despite the calibration curves also validation parameters (background equivalent concentration-BEC, detection limit-DL, internal standard-ISTD, calibration equation with correlation coefficient-R). Moreover, we added information about ICP commercial analytical standards were purchased from Agilent Technologies,  Santa Clara, CA, US (Multi-Element Calibration Standard 2A-Hg, Environmental Calibration Standard, Multi-Element Calibration Standard 2A), Merck Millipore, Darmstadt, Germany (ICP- Multi-Element Calibration Standard XVII, ICP- Multi-Element Calibration Standard VI, Phosphorus ICP standard), Honeywell Fluka™ analytical standards (Platinum Standard for ICP, Palladium Standard for ICP), and Inorganic Ventures, Christiansburg, Virginia, US (Rare Earth, Standards).

Reviewer 2 Report

The paper of Baj et al. describes an interesting work on metal analysis in brain samples and their statistical comparison between a suicide case group and a control group. Metal analysis was done by ICP-MS, which is state of the art in precise and sensitive metal quantification.

The authors should address the following comments.

It is not fully clear why all these metals were chosen for statistical comparison. What is the concept behind this list of metals? On the other hand, one important trace metal – lithium – is missing as lithium is known to be connected to psychological disorders. It is recommended that Li analysis and data should be added to the study, and to the discussion.

The conclusions are based on statistical treatment of the data (correlations). Although the mechanistic of disturbed metal homeostasis has been discussed, it is difficult to relate this only to the statistical data without experiments investigating the molecular mechanisms (causalities).

Minor:

The authors were using inductively coupled plasma mass spectrometry –ICP-MS). However, in the abstract they mention optical absorption spectrometry with inductively coupled plasma. This should be corrected in the abstract.

Author Response

The paper of Baj et al. describes an interesting work on metal analysis in brain samples and their statistical comparison between a suicide case group and a control group. Metal analysis was done by ICP-MS, which is state of the art in precise and sensitive metal quantification. The authors should address the following comments.

All authors are very grateful for the comprehensive review and insightful analysis of our work. The reviewer's suggestions have been taken into account to prepare the revised version of the manuscript. This helped us to re-think our work again.

It is not fully clear why all these metals were chosen for statistical comparison. What is the concept behind this list of metals? On the other hand, one important trace metal – lithium – is missing as lithium is known to be connected to psychological disorders. It is recommended that Li analysis and data should be added to the study, and to the discussion.

Lithium was not detected in the meninges samples. We presented all metals detected in the examined samples above the detection limits characteristic for the ICP-MS method.

The conclusions are based on statistical treatment of the data (correlations). Although the mechanistic of disturbed metal homeostasis has been discussed, it is difficult to relate this only to the statistical data without experiments investigating the molecular mechanisms (causalities).

This reviewer's comment concerns the conclusion part. Since we did not study molecular mechanisms of trace elements distribution it is difficult to give a proper conclusion. This subject area  (human meninges) is rather new so we are worrying about giving any statements about mentioned mechanisms without further investigations.

Minor: The authors were using inductively coupled plasma mass spectrometry –ICP-MS). However, in the abstract they mention optical absorption spectrometry with inductively coupled plasma. This should be corrected in the abstract.

Yes it was corrected. It was our mistake. 

Reviewer 3 Report

The paper titled “ICP-MS Multi-Elemental Analysis of the Human Meninges Collected from Sudden Death Victims in South-Eastern Poland”, is an excellent study. Another unique aspect of this study is that there are not many studies out there investigating trace elements from sudden death victims. It’s not often researchers have the chance to look analyze trace elements of the human brain, especially with such a large number of subjects n=40.

My suggestions are to increase the resolution on figure 1, this is a difficult figure to read with the current resolution. I can’t read the elements on the axis.

Another suggestion that might help the authors out and to help the data to be more readable to the viewership would be to include a principal components analysis of the data.

Figure 2 can and should be cleaned up.

In the ICP- Measurements section, how many standards were used for each element? This needs to be included here.

In the discussion the authors mention that heavy metals are found in the environment, urbanized areas as well as agriculture. These metals also are found in what people wear such as facial masks:

recently there is a relevant article regarding the leaching of trace elements from face masks into saliva. This is relevant as these elements could be absorbed into the human body:

The following article would be appropriately suited here in the discussion.

Bussan, Derek D. et al. "Quantification of trace elements in surgical and KN95 face masks widely used during the SARS-COVID-19 pandemic." Science of The Total Environment (2021): 151924.

If the authors are amenable to my suggested changes, I would accept the article with minor corrections

Author Response

The paper titled “ICP-MS Multi-Elemental Analysis of the Human Meninges Collected from Sudden Death Victims in South-Eastern Poland”, is an excellent study. Another unique aspect of this study is that there are not many studies out there investigating trace elements from sudden death victims. It’s not often researchers have the chance to look analyze trace elements of the human brain, especially with such a large number of subjects n=40.

 Thank you very much  for this opinion. All authors are very grateful for the comprehensive review and insightful analysis of our work. The reviewer's suggestions have been taken into account to prepare the revised version of the manuscript. This helped us to re-think our work again.

My suggestions are to increase the resolution on figure 1, this is a difficult figure to read with the current resolution. I can’t read the elements on the axis.

The figure was corrected.

Another suggestion that might help the authors out and to help the data to be more readable to the viewership would be to include a principal components analysis of the data.

Thank you for this suggestion. According to the reviewer suggestion we performed PCA analysis. We added the following:

Owing to the packages available in R, it is possible to perform the principal component analysis (PCA) and their visualization. Graphical PCA analysis enables the representation of an n-dimensional data set in a two-dimensional space. PCA analysis was performed for the ICP-MS data obtained for the dura mater and arachnoid mater (Figure 2). Although the PCA graph relating to the first two components explains about 40% of the variance, it allows for the separation of the imposed categories (suicide/death in an accident). As can be seen in Figure 2, in the case of the dura mater, the category fields overlap almost completely, and in the case of the arachnoid meninges, they are clearly separated.

(a)

(b)

Figure 2. Principal component analysis (PCA) biplots of ICP-MS data from the dura matter (a), and the arachnoid mater (B). Only the first two principal components (PC1, PC2) are shown with their respective variation in percentage. 

Figure 2 can and should be cleaned up.

 The figure was deleted and PCA biplots were inserted instead.

In the ICP- Measurements section, how many standards were used for each element? This needs to be included here.

The standards have been added. ICP commercial analytical standards were purchased from Agilent Technologies,  Santa Clara, CA, US (Multi Element Calibration Standard 2A-Hg, Environmental Calibration Standard, Multi Element Calibration Standard 2A), Merck Millipore, Darmstadt, Germany (ICP- Multi Element Calibration Standard XVII, ICP- Multi Element Calibration Standard VI, Phosphorus ICP standard), Honeywell Fluka™ analytical standards (Platinum Standard for ICP, Palladium Standard for ICP), and Inorganic Ventures, Christiansburg, Virginia, US (Rare Earth, Standards).

In the discussion the authors mention that heavy metals are found in the environment, urbanized areas as well as agriculture. These metals also are found in what people wear such as facial masks: recently there is a relevant article regarding the leaching of trace elements from face masks into saliva. This is relevant as these elements could be absorbed into the human body: The following article would be appropriately suited here in the discussion. Bussan, Derek D. et al. "Quantification of trace elements in surgical and KN95 face masks widely used during the SARS-COVID-19 pandemic." Science of The Total Environment (2021): 151924.

Yes. We agree with the reviewer. We added the following ref. to our discussion: Bussan, D.D.;  Snaychuk, L.; Bartzas, G.; Douvris, C. Quantification of trace elements in surgical and KN95 face masks widely used during the SARS-COVID-19 pandemic. Science of The Total Environment 2022, 814, 151924. doi: 10.1016/j.scitotenv.2021.151924.

Thank you for all suggestions.

If the authors are amenable to my suggested changes, I would accept the article with minor corrections

We do hope that our answers and improvement are appropriate and the new version of the manuscript will be accepted for publication in Molecules.

Round 2

Reviewer 1 Report

I thank the authors for partially addressing my comments. However, there is still some unclarity for me.

1) Figure 1 (Correlation matrix) is much better now but it is still blurry and the symbols of the elements can be hardly read, I suggest enhancing the resolution and changing font color to black for the element symbols.

2) I still don't follow what was the principle of selecting a p-value test. In most, different cases different statistical tools should show the same statistical trend and in the most orthodox way of interpreting p-values, its exact magnitude is actually almost not relevant, rather than fitting the cutoff to accept or abandon the null hypothesis. Were the normalities tested?

3) Figure rounding of concentrations in table 1 does not fit the actual accuracy of the method used and thus does not make any sense. With the recoveries claimed, for Na, K & Rb (the levels of ca. 400-600 ppm) it should be just a whole number, for the levels of ca. 1-5 ppm (Cs, Co etc.) I would present two decimals (actually same for those < 1). For Tb, I believe it should be only one significant figure due to low level and high uncertainty.

4) In line with item 3, avoid ppm, these are not recommended units in most cases since they can be unambiguous, use µg/g, mg/g etc. instead

Author Response

I thank the authors for partially addressing my comments. However, there is still some unclarity for me.

Answer: The authors thank the reviewer very much for his opinion. We are pleased that some of our corrections are sufficient. We will make further efforts to correct our manuscript. Once again, we would like to thank the Reviewer for helpful suggestions.

1) Figure 1 (Correlation matrix) is much better now but it is still blurry and the symbols of the elements can be hardly read, I suggest enhancing the resolution and changing font color to black for the element symbols.

Answer: We agree with the reviewer. We tried, as suggested by the reviewer, to change the font to improve the readability of the chart, but it did not help much, because there are too many symbols. Therefore, we decided to remove the symbols from the chart and list them in the figure caption.

2) I still don't follow what was the principle of selecting a p-value test. In most, different cases different statistical tools should show the same statistical trend and in the most orthodox way of interpreting p-values, its exact magnitude is actually almost not relevant, rather than fitting the cutoff to accept or abandon the null hypothesis. Were the normalities tested?

Answer: For testing normality, we used the Shapiro-Wilks test, the Anderson-Darling test, and the Lilliefors test. Because all of the observations were not present normal distribution, we used non-parametric tests only. In order to compare two or more statistically independent observations, we used the Wilcox Mann-Whitney rank-sum test and the Brunner-Munzel test, whereas, for testing statistically dependent observations, the Friedman test was used. In scientific studies, the probability level p = 0.05 is conventionally assumed as the threshold of statistical significance of the results. When the significance for a given test is lower, the results are statistically significant, while higher - insignificant. Thus, the assumed uncertainty as to the truth of the conclusion is in the order of 5%. Accordingly to above, in the interpretation of the statistical tests, a threshold of p = 0.05 was used to answer the question of whether the observed differences between the groups were statistically significant (p <0.05) or whether the differences were a matter of chance (p>0.05). According to the reviewer's suggestion, we simplified subchapter 2.2.

3) Figure rounding of concentrations in table 1 does not fit the actual accuracy of the method used and thus does not make any sense. With the recoveries claimed, for Na, K & Rb (the levels of ca. 400-600 ppm) it should be just a whole number, for the levels of ca. 1-5 ppm (Cs, Co etc.) I would present two decimals (actually the same for those < 1). For Tb, I believe it should be only one significant figure due to the low level and high uncertainty.

Answer: Thank You for this suggestion. The values have been corrected according to the above suggestion.

4) In line with item 3, avoid ppm, these are not recommended units in most cases since they can be unambiguous, use µg/g, mg/g etc. instead

Answer: The abbreviation has been explained under table 1. Once again all authors appreciate the reviewer’s suggestions and any efforts made in the aim to improve our manuscript.

Reviewer 2 Report

The authors answered all my comments and improved the paper.

The strong point is that the investigated subject (metal analysis in human meninges) is original.

The major weak point is still, with respect to the answers of the authors, that the rationale of the study is methodological driven (the elements which can be detected are relevant). Thus, the study is rather observational with statistical data treatment than an investigation of the biological role and action of the regarded elements. However, the presented data could be a starting point for further studies. 

Author Response

The authors answered all my comments and improved the paper.

The strong point is that the investigated subject (metal analysis in human meninges) is original.

The major weak point is still, with respect to the answers of the authors, that the rationale of the study is methodological driven (the elements which can be detected are relevant). Thus, the study is rather observational with statistical data treatment than an investigation of the biological role and action of the regarded elements. However, the presented data could be a starting point for further studies. 

Answer: Thank you very much for this opinion. We fully agree with this assessment. Human meningeal tissues have not been subjected to any analysis so far, so our analysis is the first in this research area. The work is indeed not manipulative, but rather observational (as most studies in the field of medicine due to the ethics of human research) in terms of designing an experiment, but the results of statistical analyzes may be useful to physicians, especially psychiatrists, and inspire further research. To clarify the perspectives of our work we added the following sentence to the conclusion: Since the disturbing metals' homeostasis is related to diverse neuropathologies and behavioral dysfunctions, our observations with statistical data treatment could be the starting point for further studies to physicians, especially psychiatrists.”